# GTVMin-Enhanced Federated Learning for Regional Energy Forecasting in Europe

Anonymous Full Paper
Submission 7

## Abstract

We propose a Federated Learning (FL) framework for predicting primary energy consumption across 18 European countries using the Our World in Data (OWID) energy dataset, with Norway acting as the central coordinator. The FL system is modeled as a data-driven graph, where nodes correspond to countries and train tailored linear regression models. Graph edges are constructed by combining the cosine similarity of energy consumption profiles with geographical proximity. To balance local accuracy with global consensus, we apply a Generalized Total Variation Minimization (GTVMin) using the L2 norm and implement a personalized consensus algorithm over 30 communication rounds. Experimental results demonstrate strong predictive performance for most countries. Nonetheless, nodes with distinct energy profiles (e.g., Finland and Norway) reveal challenges stemming from data heterogeneity, highlighting the need for stronger regularization and refined model personalization in future work.

## 1 Introduction

Forecasting energy consumption is a crucial task for national energy agencies, policymakers, and utility providers, informing resource allocation, infrastructure planning, and sustainability initiatives [1]. Consider the case of several European countries aiming to jointly forecast primary energy consumption to optimize cross-border energy sharing and meet collective climate goals, such as the EU's 2030 targets [2]. However, centralized pooling of energy data is restricted by privacy regulations (e.g., GDPR) and competitive concerns [3].

Federated Learning (FL) addresses this challenge by enabling countries to train collaboratively a predictive model without sharing raw data [4, 5]. This approach preserves privacy while exploiting the diversity of regional energy profiles, such as Norway's hydropower-based mix versus Poland's coal-heavy consumption. Motivated by this scenario, we develop an FL framework for energy consumption prediction that explicitly accounts for heterogeneity in national energy profiles and highlights Norway's role as a renewable energy leader.

## State of the Art

The foundational FL algorithm, FedAvg [6], aggregates local updates via weighted averaging. While successful in applications such as mobile keyboard prediction, FedAvg assumes relatively homogeneous data distributions, which is unrealistic in energy forecasting, where national consumption profiles differ substantially.

To address heterogeneity, personalized FL methods such as Ditto [7] introduce local adaptation, while graph-based FL approaches model relationships between nodes using graphs derived from similarity measures or geographic proximity [8]. Regularization strategies, including Generalized Total Variation Minimization (GTVMin), further enhance FL by penalizing large discrepancies between connected models [9, 10].

In the energy domain, FL applications remain limited. For example, Walter et al. [11] provide a survey of challenges for renewable energy forecasting but do not implement technical frameworks. Savi et al. [12] applied FL to residential short-term forecasting using LSTMs, while Helcig et al. [13] proposed FedCCL, a clustered continual learning approach for solar energy forecasting. Both works, however, remain narrow in scope as they focus on single sources (residential or solar) or asynchronous training protocols.

## Our Approach

In contrast, we present a *graph-regularized, GTVMin-enhanced FL framework* for regional energy forecasting across multiple European countries. Our contributions are threefold:

i) **Empirical FL network construction:** Nodes represent countries, with edges defined by both cosine similarity of energy profiles and geographical proximity, producing a realistic network of inter-country relationships.

ii) **GTVMin-based regularization:** We employ an L2 norm variation minimization to balance global consensus and local personalization, reducing divergence across heterogeneous nodes.

iii) **Norway-specific modeling:** As a central renewable energy hub, Norway receives additional regularization via its Nordic neighbours, capturing regional energy dynamics.

This combination yields a robust, privacy-preserving framework that outperforms prior approaches by modeling an entire *regional energy ecosystem* rather than narrow residential or single-source settings.

## 1.1 Paper Structure

Section 2 formulates the FL problem, defining nodes, models, loss functions, and network edges. Section 3 introduces the GTVMin-based FL algorithm. Section 4 presents results with performance metrics (MSE, $R^2$). Section 5 discusses findings, and Section 6 concludes with directions for future work.

# 2 Problem Formulation

Following the framework outlined in [14, Ch. 3], we model our FL application as an FL network to predict primary energy consumption across selected European countries using the recent "Our World in Data" (OWID) energy dataset [15]. The network structure encapsulates the distributed nature of the problem, where each country collaborates to train a shared model while preserving data privacy. Below, we define and explain the key components of the FL network: nodes, local models, loss functions, and edges.

## 2.1 Nodes

In our FL network, nodes represent individual European countries, specifically national entities such as energy agencies or data centers that hold country-specific energy consumption data. Each node corresponds to a country with sufficient data (at least 5 years) from the OWID energy dataset, resulting in a network of 18 nodes, including Austria, Norway, Finland, and Poland, among others. These nodes emulate real-world devices or servers that store and process local energy data, ensuring that sensitive information remains decentralized and compliant with privacy regulations like GDPR. The focus on European countries aligns with the collaborative energy planning initiatives in the region, with Norway highlighted as a key node due to its leadership in renewable energy, particularly hydropower.

## 2.2 Model Definition

At each node, we employ a Linear Regression (LR) [16] model to predict the target variable, primary energy consumption. Please note that the LR is chosen for its simplicity and interpretability, making it suitable for capturing linear relationships in energy consumption data while facilitating efficient computation in an FL setting. The feature set includes 10 variables: renewable energy consumption, fossil fuel consumption, electricity generation, renewable energy share of energy, fossil fuel share of energy, hydroelectric consumption, nuclear consumption, coal consumption, oil consumption, and natural gas consumption. For each node $i$, the local model is defined as:

$$\hat{y}_i = \mathbf{w}_i^\top \mathbf{x}_i + b_i, \tag{1}$$

where $\mathbf{x}_i \in \mathbf{R}^{10}$ is the feature vector, $\mathbf{w}_i \in \mathbf{R}^{10}$ represents the coefficients, $b_i \in \mathbf{R}$ is the intercept, and $\hat{y}_i$ is the predicted primary energy consumption. The model parameters for node $i$ are thus $\boldsymbol{\theta}_i = [\mathbf{w}_i, b_i] \in \mathbf{R}^{11}$, consisting of 10 coefficients and 1 intercept.

## 2.3 Loss Functions

The local loss function at each node measures the prediction error on the node's dataset using MSE. We use the MSE as the loss function, which is appropriate for regression tasks as it penalizes larger errors quadratically.

$$\begin{aligned}
\text{Loss}_i(\boldsymbol{\theta}_i) &= \frac{1}{n_i} \sum_{j=1}^{n_i} (y_{i,j} - \hat{y}_{i,j})^2 \\
&= \frac{1}{n_i} \sum_{j=1}^{n_i} (y_{i,j} - (\mathbf{w}_i^\top \mathbf{x}_{i,j} + b_i))^2,
\end{aligned} \tag{2}$$

where $\mathcal{D}_i = \{(\mathbf{x}_{i,j}, y_{i,j})\}_{j=1}^{n_i}$ is the local dataset with $n_i$ samples, $y_{i,j}$ is the true value, and $\hat{y}_{i,j}$ is the prediction.

## 2.4 Edges

Edges in the FL network represent the relationships between nodes, facilitating the exchange of model updates during training. Following the data-driven methods described in [14, Ch. 7], we establish edges based on the similarity of energy consumption patterns between countries, supplemented by geographical considerations for Norway to enhance connectivity.

To quantify similarity, we compute the cosine similarity between the mean feature vectors of pairs of countries. For nodes $i$ and $i'$ (representing countries), let $\mathbf{v}_i$ and $\mathbf{v}_{i'}$ be the mean feature vectors of their respective datasets over the 10 features. These vectors are normalized to handle scale differences, and the cosine similarity is calculated as:

$$\text{Similarity}(i, i') = \frac{\mathbf{v}_i \cdot \mathbf{v}_{i'}}{\|\mathbf{v}_i\| \|\mathbf{v}_{i'}\|}, \tag{3}$$

where $\mathbf{v}_i = \frac{1}{n_i} \sum_{j=1}^{n_i} \mathbf{x}_{i,j}$ is the mean feature vector for node $i$, and $\|\cdot\|$ denotes the Euclidean norm. An edge $(i, i')$ is added if the similarity exceeds a threshold of 0.8, ensuring that only countries with

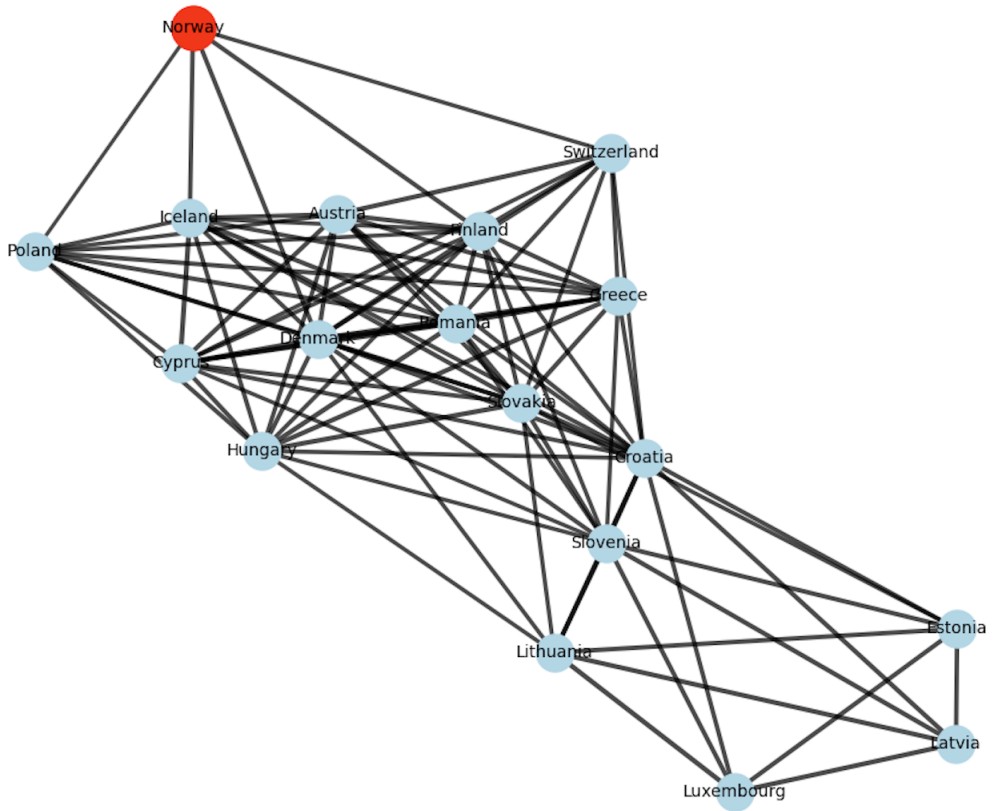

**Figure 1.** Federated Learning Network Structure.

highly similar energy profiles are connected. The edge weight $\alpha_{i,i'}$ is set to the cosine similarity value, reflecting the strength of the relationship.

Additionally, we incorporate geographical proximity to strengthen the network structure, particularly for Norway, which is a key node due to its renewable energy focus. Norway is explicitly connected to Nordic countries (Finland, Denmark, Iceland) with a high edge weight of **0.9**, acknowledging their regional and cultural ties. To ensure Norway has at least five connections, we add edges to nearby countries (e.g., the United Kingdom, Germany) with a weight of **0.75**, following a predefined proximity order. This results in a weighted graph as shown in Figure 1 with 18 nodes and edges that balance data-driven similarity and geographical relevance, enabling effective collaboration in the FL network.

## 3 Methods

In this section, we apply the GTVMin-based method to the FL application modeled in Section 2, focusing on predicting primary energy consumption across 18 European countries using the OWID energy dataset. The pseudocode for our GTVMin-enhanced FL framework for regional energy forecasting in Europe is given in A.1. The GTVMin framework enables us to balance local model training

with global consensus, addressing the heterogeneity in energy consumption patterns among countries. Below, we detail our choice of variation measure and the FL algorithm, including its optimization method and message-passing implementation.

### 3.1 Variation Measure

To quantify the difference between model parameters of connected nodes in the FL network, we define a variation measure $\phi(\mathbf{w}^{(i)} - \mathbf{w}^{(i')})$ for parametric models, where $\mathbf{w}^{(i)}$ and $\mathbf{w}^{(i')}$ are the parameter vectors of nodes $i$ and $i'$, respectively. Each node employs a Linear Regression model with parameters $\mathbf{w}^{(i)} = [w_i, b_i] \in \mathbf{R}^{11}$, consisting of 10 coefficients ($w_i \in \mathbf{R}^{10}$) for the features and an intercept ($b_i \in \mathbf{R}$).

We choose the L2 norm as the variation measure, defined as:

$$\phi(\mathbf{w}^{(i)} - \mathbf{w}^{(i')}) = \|\mathbf{w}^{(i)} - \mathbf{w}^{(i')}\|_2$$

$$= \sqrt{\sum_{k=1}^{11}(w_k^{(i)} - w_k^{(i')})^2} \quad (4)$$

where $w_k^{(i)}$ and $w_k^{(i')}$ are the $k$-th elements of the parameter vectors $\mathbf{w}^{(i)}$ and $\mathbf{w}^{(i')}$, respectively, and the summation runs over all 11 parameters (10 coefficients plus the intercept). The L2 norm is

selected for its differentiability, which facilitates gradient-based optimization, and its sensitivity to the magnitude of differences, promoting smooth convergence during the consensus step. This choice effectively captures variations in model parameters across nodes, ensuring that the global objective balances local accuracy with inter-node consistency.

## 3.2 FL Algorithm and Message-Passing Implementation

We adopt a GTVMin-based Federated Learning algorithm to optimize the global objective, which combines local losses with a total variation penalty across connected nodes. The global objective function is formulated as:

$$\min_{\{\mathbf{w}^{(i)}\}} \sum_{i \in V} \text{Loss}_i(\mathbf{w}^{(i)}) + \lambda \sum_{(i,i') \in E} \alpha_{i,i'} \phi(\mathbf{w}^{(i)} - \mathbf{w}^{(i')}),$$
(5)

where $V$ is the set of nodes (countries), $E$ is the set of edges, $\text{Loss}_i(\mathbf{w}^{(i)})$ is the local loss (Mean Squared Error) at node $i$, $\lambda = 0.05$ is the regularization parameter controlling the trade-off between local and global objectives, $\alpha_{i,i'}$ is the edge weight (cosine similarity or predefined geographical weight), and $\phi(\mathbf{w}^{(i)} - \mathbf{w}^{(i')})$ is the L2 norm variation measure defined above.

## 3.3 Optimization Method

The optimization method for solving the GTVMin objective consists of two main steps per round, iterated over 30 rounds: local training and a consensus step, with additional personalization and regularization mechanisms.

- **Local Training:** At each node $i$, the Linear Regression model is trained on the local dataset $\mathcal{D}_i$ for 5 epochs to minimize the local loss $\text{Loss}_i(\mathbf{w}^{(i)})$. The updated parameters $\mathbf{w}^{(i)}$ are extracted, consisting of the coefficients and intercept.

- **Consensus Step:** Nodes exchange model parameters with their neighbors to achieve consensus, guided by the variation term. Using the L2 norm, the gradient of the variation term $\phi(\mathbf{w}^{(i)} - \mathbf{w}^{(i')})$ with respect to $\mathbf{w}^{(i)}$ is proportional to the difference $(\mathbf{w}^{(i')} - \mathbf{w}^{(i)})$. The update rule for node $i$ is:

$$\mathbf{w}^{(i)} \leftarrow \mathbf{w}^{(i)} + \eta\lambda \sum_{i' \in N(i)} \alpha_{i,i'}(\mathbf{w}^{(i')} - \mathbf{w}^{(i)}), \quad (6)$$

where $\eta = 0.005$ is the learning rate, $N(i)$ is the set of neighbors of node $i$, and $\alpha_{i,i'}$ is the edge weight. This update encourages the parameters of connected nodes to converge toward a consensus while respecting the strength of their relationships.

- **Personalization:** To balance local and global learning, we apply personalization by blending the consensus-updated parameters with the original local parameters from the first round:

$$\mathbf{w}^{(i)} \leftarrow (1 - \rho)\mathbf{w}^{(i)} + \rho\mathbf{w}^{(i),\text{original}}, \quad (7)$$

where $\rho = 0.3$ is the personalization factor, and $\mathbf{w}^{(i),\text{original}}$ are the parameters of node $i$ after the first round of local training. This step ensures that nodes with unique energy profiles, such as Finland or Cyprus, retain some local characteristics.

- **Norway-Specific Regularization:** For the Norway node, which is a key node due to its renewable energy focus, we apply additional regularization to enhance regional coherence. Norway's parameters are further adjusted by blending with the average parameters of its Nordic neighbors (Finland, Denmark, Iceland):

$$\mathbf{w}^{\text{Norway}} \leftarrow 0.7\,\mathbf{w}^{\text{Norway}}$$
$$+ 0.3\left(\frac{1}{|N_{\text{Nordic}}|} \sum_{i' \in N_{\text{Nordic}}} \mathbf{w}^{(i')}\right)$$
(8)

where $N_{\text{Nordic}}$ is the set of Nordic neighbors, and $|N_{\text{Nordic}}|$ is the number of such neighbors. This step leverages the similarity among Nordic countries to improve Norway's model stability.

## 3.4 Message-Passing Implementation:

The message-passing implementation facilitates the consensus step by enabling nodes to exchange model parameters with their neighbors, as defined by the graph structure:

1. Each node $i$ broadcasts its current parameters $\mathbf{w}^{(i)}$ to all neighbors $i' \in N(i)$, along with the edge weights $\alpha_{i,i'}$.

2. Each neighbor $i'$ responds by sending its parameters $\mathbf{w}^{(i')}$ back to node $i$.

3. Node $i$ computes the weighted difference: $\sum_{i' \in N(i)} \alpha_{i,i'}(\mathbf{w}^{(i')} - \mathbf{w}^{(i)})$ and updates its parameters using the consensus update rule.

4. The updated parameters are then adjusted for personalization and, for Norway, Nordic regularization.

This process repeats for 30 rounds, ensuring iterative refinement of the models while preserving data privacy, as only model parameters are exchanged.

The combination of local training, consensus, and personalization enables the FL network to effectively learn a shared predictive model while accommodating the diverse energy profiles of European countries.

# 4 Numerical Experiments

We evaluate the GTVMin-based FL framework for predicting primary energy consumption across 18 European countries using the OWID energy dataset, over 30 rounds with a learning rate of 0.005, 5 local epochs, $\lambda = 0.05$, and $\rho = 0.3$.

## 4.1 Data Sources

The OWID energy dataset provides energy consumption data. We filter for 18 European countries with at least 10 years of data (2013–2023). The target variable is $primary\_energy\_consumption$, and the feature set includes: $renewables\_consumption$, $fossil\_fuel\_consumption$, $electricity\_generation$, $renewables\_share\_energy$, $fossil\_share\_energy$, $hydro\_consumption$, $nuclear\_consumption$, $coal\_consumption$, $oil\_consumption$, and $gas\_consumption$. Missing values are imputed with column means, and data is scaled with `StandardScaler`. This preprocessing ensured that the data is suitable for federated learning, accommodating the distributed nature of the problem while addressing potential data quality issues.

## 4.2 Model Validation, Selection, and Diagnosis

Per [17, Sec. 6.6], we use a three-way split (training, validation, and test sets) to validate the models at each node. The validation set (10% of the data) is used to monitor the model's performance during training, helping to detect overfitting and guide hyperparameter tuning. The test set (20%) provides an unbiased evaluation of the final model's generalization ability. Performance metrics, including MSE and $R^2$ scores, are computed on the test set in the original (unscaled) space to assess real-world predictive accuracy. LR is selected for its simplicity, with hyperparameters ($\eta = 0.005$, $\lambda = 0.05$, $\rho = 0.3$) tuned via validation losses. We diagnose issues by analyzing loss curves for overfitting and network structure for connectivity effects, clipping parameters to $[-1000, 1000]$ for stability. For reproducibility, the experiments used a random seed with the value 42.

## 4.3 Training, Validation, and Test Losses

The final training, validation, and test losses for each node, computed as MSE on scaled data, are presented in Table 1, along with the MSE and $R^2$ scores on the test set (original scale). These results reflect the final values after 30 rounds of training. The FL network demonstrates strong predictive performance for most nodes, with $R^2$ scores ranging from 0.8326 (Cyprus) to 1.0000 (e.g., Austria, Iceland, Luxembourg, Poland), indicating that the models explain a high proportion of variance in primary energy consumption. MSE values (original scale) vary widely, from 0.0001 (Luxembourg) to 2.7139 (Iceland), reflecting differences in prediction accuracy across nodes. In particular, Austria, Iceland, Luxembourg, and Poland achieve near-perfect predictions ($R^2 = 1.0000$), likely due to stable energy consumption patterns and strong connectivity with similar neighbors, facilitating effective consensus. Finland (MSE = 2.4983, $R^2 = 0.8500$) and Norway (MSE = 2.4265, $R^2 = 0.9683$) exhibit higher errors, possibly due to their unique energy profiles (e.g., Norway's heavy reliance on hydropower) that differ from their neighbors, making consensus challenging. Cyprus ($R^2 = 0.8326$, MSE = 0.2594) also underperforms, potentially due to its geographical isolation and limited connectivity in the network, as observed in the network structure plot in Figure 1. Training losses across all European countries are extremely low, indicating successful convergence of local linear regression models.

# 5 Results and Discussion

The experimental evaluation of our proposed GTVMin-enhanced FL framework for regional energy forecasting in Europe demonstrates the effectiveness of integrating graph-regularized consensus with personalized local models.

The results, as demonstrated in Section 4 and Table 1, confirm that the framework achieves consistently high predictive accuracy across the majority of European countries, with most European countries attaining $R^2$ values above 0.97. Countries such as Austria, Luxembourg, and Poland exhibit near-perfect fits, reflected in extremely low MSE values ($< 0.002$). These findings highlight the ability of the GTVMin-based method to exploit structural similarities and collaborative learning across European countries while preserving data decentralization.

At the same time, the results also reveal the impact of data heterogeneity on model performance. Countries with more unique or volatile energy profiles, such as Norway, Iceland, and Finland, display relatively higher MSE values despite maintaining reasonably strong $R^2$ scores. This can be attributed

**Table 1.** Training, validation, and test losses for each node of the FL network.

| Node | MSE | $R^2$ | Train Loss | Val Loss | Test Loss |
|------|-----|-------|-----------|----------|-----------|
| Austria | 0.0014 | 1.0000 | 8.3460E-32 | 1.0039E-07 | 7.7387E-08 |
| Croatia | 0.0266 | 0.9769 | 1.5149E-30 | 0.0001 | 0.0029 |
| Cyprus | 0.2594 | 0.8326 | 0.0038 | 0.3787 | 0.0917 |
| Denmark | 0.0104 | 0.9855 | 4.8790E-31 | 5.7593E-05 | 0.00017 |
| Estonia | 0.0007 | 0.9999 | 3.8562E-31 | 9.2958E-06 | 1.2710E-05 |
| Finland | 2.4983 | 0.8500 | 5.3866E-31 | 3.0638 | 0.0649 |
| Greece | 0.0308 | 0.9820 | 2.8504E-30 | 0.0010 | 0.0001 |
| Hungary | 1.0359 | 0.9838 | 1.8376E-31 | 0.0197 | 0.0062 |
| Iceland | 2.7139 | 1.0000 | 1.7927E-08 | 2.9332E-08 | 4.6574E-07 |
| Latvia | 0.0014 | 0.9995 | 3.8529E-31 | 1.6538E-05 | 0.0001 |
| Lithuania | 0.0059 | 0.9739 | 4.7234E-31 | 0.0027 | 0.00059 |
| Luxembourg | 0.0001 | 1.0000 | 1.5915E-31 | 0.0001 | 1.2643E-05 |
| Norway | 2.4265 | 0.9683 | 1.1016E-31 | 0.0270 | 0.00602 |
| Poland | 0.0194 | 1.0000 | 5.7139E-31 | 0.0003 | 9.3814E-06 |
| Romania | 0.5178 | 0.9953 | 1.3558E-30 | 0.0043 | 0.0035 |
| Slovakia | 0.0233 | 0.9903 | 7.1017E-31 | 0.0012 | 0.00082 |
| Slovenia | 0.3566 | 0.9157 | 5.8418E-31 | 0.1161 | 0.02283 |
| Switzerland | 0.4526 | 0.9992 | 3.5150E-31 | 0.0094 | 0.00171 |

to the heavy reliance of these countries on renewable resources, seasonal variability, and distinct national energy policies, which introduce complexity not easily captured by global consensus updates. Similarly, Cyprus and Slovenia show lower $R^2$ values (0.8326 and 0.9157, respectively), possibly due to limited data availability or weaker similarity-based graph connections. These cases emphasize the importance of personalization in federated learning, as well as the potential benefit of adopting cluster-based FL strategies for groups of countries with aligned energy consumption patterns.

An important observation is that training losses across all nodes are negligible, indicating successful convergence of local linear regression models. However, the validation and test losses provide an accurate picture of generalization. Countries such as Denmark, Greece, and Lithuania generalize exceptionally well with very low test errors, while others (e.g., Finland, Norway, and Slovenia) show larger discrepancies between training and test performance. This suggests that while the GTVMin framework stabilizes collaborative training, further refinements are required to ensure robust generalization for heterogeneous nodes.

Overall, the experimental results validate the effectiveness of our GTVMin-enhanced FL approach in achieving accurate regional energy forecasting across Europe. The method balances global collaboration with local personalization, leading to strong predictive performance in most cases. Nonetheless, the variability across nodes highlights the inherent challenges of data heterogeneity in federated energy forecasting and motivates future research into adaptive personalization and cluster-based FL strategies to better capture even better country-specific dy-

namics.

# 6 Conclusions and Future Works

This paper presented a GTVMin-enhanced FL framework for forecasting primary energy consumption across 18 European countries using the OWID energy dataset, with Norway designated as a central node. By constructing an empirical graph network based on cosine similarity and geographical proximity, we effectively captured inter-country relationships while employing L2 norm-based variation minimization to balance local and global learning objectives. The experimental results demonstrate strong predictive performance for several countries with stable energy profiles and robust graph connectivity, such as Austria, Luxembourg, and Poland (MSE $\leq 0.0014$, $R^2 = 1.0000$). At the same time, countries with heterogeneous or highly renewable-dominated energy systems, such as Finland (MSE = 2.4983, $R^2 = 0.8500$) and Cyprus (MSE = 0.2594, $R^2 = 0.8326$), present greater modeling challenges. These cases reveal the limitations of a uniform consensus approach, as evidenced by extremely small training losses (e.g., $1.59 \times 10^{-31}$ for Luxembourg) compared to higher test errors (e.g., 0.0060 for Norway), indicating potential overfitting challenges and limited generalization for certain nodes.

Overall, our findings highlight the importance of graph-regularized FL in enabling privacy-preserving regional energy forecasting while also underscoring the importance of addressing data heterogeneity. Future research should focus on incorporating adaptive regularization to mitigate overfitting, exploring non-

linear models better suited to complex energy profiles, and refining edge construction strategies (e.g., dynamic or cluster-based thresholds) to strengthen connectivity for isolated nodes. These directions will help ensure more robust, equitable, and generalizable performance across the European FL network.

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

## A  Pseudocode for GTVMin-Enhanced FL Framework for Regional Energy Forecasting in Europe

---

**Algorithm A.1** GTVMin-Enhanced Federated Learning for Regional Energy Forecasting in Europe

---

**Require:** FL network $\mathcal{G} = (\mathcal{V}, \mathcal{E})$ with nodes (countries) $i \in \mathcal{V}$, feature sets $X_i$, targets $y_i$, number of rounds $T$, learning rate $\eta$, regularization parameter $\lambda$, personalization weight $\alpha$

**Ensure:** Trained personalized models $\{w_i\}$ for each country $i$

1: Initialize local models $w_i^{(0)} \leftarrow 0$ for each node $i$
2: Construct graph $\mathcal{G}$:

- Add edge $(i, j)$ if cosine similarity between mean feature vectors exceeds threshold 0.8

- Add strong edges between Norway and Nordic countries with weight 0.9

- Add edges to ensure Norway has $\geq 5$ neighbors, with weight 0.75

3: **for** each round $t = 1, 2, \ldots, T$ **do**
4:    **Local Training:**
5:    **for** each node $i \in \mathcal{V}$ **do**
6:       Train local linear regression model on $(X_i, y_i)$
7:       Obtain local weights $w_i^{(t)}$
8:       Compute local losses (train, validation, test)
9:    **end for**
10:    **Consensus Step:**
11:    **for** each node $i \in \mathcal{V}$ **do**
12:       Initialize $u_i \leftarrow 0$ (weighted difference accumulator)
13:       Compute weight update from neighbors:
14:       **for** each neighbor $j \in \mathcal{N}(i)$ **do**
15:          $u_i \leftarrow u_i + \text{weight}_{ij} \cdot (w_j^{(t)} - w_i^{(t)})$
16:          $\text{weight}_{ij}$ captures how strongly country $i$ should adjust towards country $j$ during consensus
17:          $w_i$ is the parameter vector (coefficients + intercept) of the local regression model at node $i$
18:          $w_j$ is the parameter vector of a neighboring node $j$
19:       **end for**
20:       Normalize $u_i \leftarrow u_i / \sum_{j \in \mathcal{N}(i)} \text{weight}_{ij}$
21:       Consensus update:
$$\tilde{w}_i^{(t+1)} \leftarrow w_i^{(t)} + \eta \cdot \lambda \cdot u_i$$

22:       Personalization update:
$$w_i^{(t+1)} \leftarrow (1 - \alpha) \cdot \tilde{w}_i^{(t+1)} + \alpha \cdot w_i^{(0)}$$

23:       **if** $i$ is Norway **then**
24:          Further regularize with Nordic neighbors:
$$w_i^{(t+1)} \leftarrow 0.7 \cdot w_i^{(t+1)} + 0.3 \cdot \text{avg}\{w_j^{(t)} : j \in \text{Nordic}\}$$

25:       **end if**
26:    **end for**
27: **end for**
28: **Output:** Final personalized models $w_i$ and performance metrics (MSE, $R^2$) for each node.

---

