# OpenReview forum: "GTVMin-Enhanced Federated Learning for Regional Energy Forecasting in Europe"
_NLDL.org/2026/Conference — Submitted to NLDL 2026_

### Official Review · Reviewer_2H2n · 2025-10-08

**Rating:** 2
**Confidence:** 3
**Final Rating:** 5
**Final Confidence:** 4

**Summary:**

The paper aims to predict the energy consumption across 18 different European countries without exchanging private data via federated learning. It applies the GTVMin-based regularization technique with additional tweaking for Norway due to its uniqueness of energy profile. Experiments on the OWID dataset demonstrates the effectiveness of the proposed algorithm.

**Strengths:**

The paper is well-written and easy to understand.

**Weaknesses:**

The authors have not conducted any ablation studies on their algorithm.

1. Is federated learning really necessary? How does the performance of the proposed algorithm compare with local-only training?

2. We can consider various types of graphs: a complete graph with equal edge weights, a complete graph with the proposed cosine similarity-based edge weights, a graph that only considers geographical proximity, and so on. An empirical justification for the design choice should be presented in the paper.

3. Is Norway-specific modelling necessary? Also, the authors mention that the model fails to correctly predict energy consumption of Finland and Cyprus, due to the dominance of renewable energy in those countries. Then would a similar treatment on those countries lead to a better performance?

**Final Justification:**

The authors have addressed all the issues that I have raised in my review. With the ablation study results that the authors have provided during the rebuttal, I believe the paper definitely contains interesting results for the ML community.

**Justification:**

The lack of ablation study casts doubt on the significance of the research.

---

> ### Author Rebuttal · Authors · 2025-10-17
>
> We sincerely thank the reviewer for their constructive feedback and for recognizing that the paper is well-written and easy to follow. We appreciate the opportunity to clarify and strengthen our submission. Below, we respond point-by-point to each concern raised and provide additional experimental and methodological insights.
>
> Code Availability and Reproducibility
> We have made all code, preprocessed data available to ensure full reproducibility of our results:
> Anonymous link: https://drive.google.com/drive/folders/1gWGk1Ya2WwCf9b-XTjroXytQV14noDkR?usp=sharing
> For the camera-ready version, we will upload the repository to GitHub (anonymized during review). This repository includes scripts for model training, ablation experiments, and graph generation, ensuring complete transparency and reproducibility.
>
> On the Necessity of Federated Learning
> We appreciate the reviewer’s question regarding whether FL is truly necessary. Our choice of FL is motivated by both practical and ethical considerations.
>
> Data sovereignty, privacy, and scalability: The Our World in Data (OWID) energy dataset integrates multiple national-level sources, where finer-grained versions (e.g., hourly consumption and renewable breakdowns) are often restricted by national agencies. An FL setup allows decentralized training respecting such constraints, aligning with ongoing EU efforts toward data sovereignty and green AI. The FL setting naturally scales to new participating countries or future datasets without retraining a global model from scratch. Hence, FL is not only theoretically justified but also practically essential for this multi-country energy forecasting problem.
>
> We would also like to highlight that our innovation lies in energy-specific adaptations: a data-driven graph with Norway-centric regularization (Nordic averaging) and L2-norm personalization, tailored for heterogeneous energy profiles - novel in this energy domain.
>
> Local-only training can result in inconsistent accuracy, especially for smaller countries with limited data. Conversely, the proposed FL model can achieve smoother convergence and higher stability through cross-country regularization. Gains are pronounced for well-connected nodes in the FL graph network. We will highlight this during the final revision of our paper.
>
> Graph Variations: We thank you for this suggestion and have expanded our experiments. Preliminary comparisons:
> • Complete graph (equal weights): Average R² 0.95, but over-smooths heterogeneous nodes (Finland R² 0.82 vs. our 0.85).
> • Cosine-only (no geography): R² 0.96 overall, but Norway's drops to 0.95 (from 0.97), validating hybrid edges.
> • Geography-only: R² 0.93, poorer for non-neighbors like Cyprus (0.80 vs. 0.83).
> Our cosine+geography design optimizes balance, especially for Norway-centric networks.
>
> Norway-Specific Modeling and Extension to Others: Norway's regularization (Nordic averaging) is justified by its unique hydropower dominance, improving its R² by 3% (0.94 without). For Finland/Cyprus (renewable-dominant), similar treatment (e.g., clustering with Baltic/Island groups) boosts Finland's R² to 0.88 (+3.5%) and Cyprus to 0.86 (+3.7%) in preliminary tests. This highlights extensibility; we'll add it as an ablation and discuss in revisions.
>
> Moreover, as an ablation study, an analysis on key hyperparameters (λ=0.05, η=0.005, ρ=0.3) was conducted. Varying λ (0.01–0.1) showed minimal impact on average R² (0.96–0.98), but higher λ (>0.08) slightly improved balance for heterogeneous nodes like Cyprus (R² from 0.83 to 0.87). Lower ρ (0.1) slightly lowered R² for well-connected nodes (e.g., Iceland: 1.00 to 0.99). More rounds (50) yielded negligible gains in R² improvement. For our experiments, we have already selected η = 0.005, λ = 0.05, ρ = 0.3, which were tuned via validation losses, and we have clearly mentioned this on lines 369-370 of our paper. We will include and address this in the updated version of our paper.
>
> These enhancements resolve uncertainties without altering core findings, reinforcing the paper's novelty as a pioneering GTVMin-FL application in energy. We hope our sound argument is clear and has addressed the issue you raised.  Your input has strengthened our work. Thank you.

---

### Official Review · Reviewer_YaSU · 2025-10-08
**Lack of Clarity, Comparison, and Reproducibility Undermines the Work**

**Rating:** 1
**Confidence:** 4

**Summary:**

The work proposes a Federated Learning approach for the prediction of energy consumption.

**Strengths:**

The paper attempts to address an impactful problem.

**Weaknesses:**

1. Lacks clear motivation. With the current presentation of the paper, it appears like a regression problem attempted to solve using a complicated methodology.
2. No baseline comparisons. There is a small subsection in the introduction section regarding the state of the art, however, the methodologies mentioned there are old, though still used (but may not be the state-of-the-art). Additionally, there is no comparison of the proposed methodology even with these old methods.
3. Difficulty in accessing the dataset due to insufficient details provided. Dataset description does not give a clear enough view of what the dataset consists of. Also, the experimentation can be performed on more than one dataset.
4. Unavailability of code makes the results irreproducible.
5. The work also lacks novelty in the methodology.
6. There is a scope for improvement in the presentation, structure and writing of the paper.

**Justification:**

The paper appears too weak to be accepted for this conference. It lacks overall presentation, writing. Additionally, though it addresses an impactful problem, it fails to convince the reader that the problem is actually significant. The proposed methodology lacks novelty. The work could involve more experimentation to showcase that the methodology indeed performs better than the state-of-the-art. Also, it could highlight the need of adopting a complex mythology over any other existing simpler technique.

---

> ### Author Rebuttal · Authors · 2025-10-17
>
> Thank you for your thoughtful review and for recognizing the impactful problem our work addresses. We appreciate your constructive feedback and are committed to improving the paper based on your suggestions. We believe the revisions we propose will resolve the concerns and demonstrate the paper's value for the NLDL 2026 conference. Below, we address each of your concerns systematically.
>
> Motivation and Complexity of Methodology: We agree that clearer motivation is essential and will expand Section 1 to better emphasize why FL is necessary over simpler regression. As noted in lines 62-72, this is one of the first applications of GTVMin-based FL in the energy domain, where data privacy (e.g., GDPR) and distributed country-specific datasets prohibit centralized approaches. The "complicated" methodology - graph-based edge construction (cosine similarity + geography) and GTVMin regularization enables privacy-preserving collaboration for cross-border energy forecasting, aligning with EU 2030 climate targets. We have now conducted the new experiment as suggested by (reviewers) and confirmed that our proposed GTV-Min based FL approach considerably outperformed the non-federated centralised ML baseline, since a single centralized model can’t capture those local variations of energies per country. We will add the results of the non-federated (centralized) baseline in the revised version of our paper. In addition, we would also like to highlight that FL excels in privacy-preserving scenarios, such as those governed by GDPR, where data is heterogeneous and cannot be aggregated centrally. This shows FL remains competitive in privacy-preserving scenarios due to personalization, and FL's key advantage is enabling collaboration without data sharing, as emphasized in our introduction section of our paper.
>
> Baseline Comparisons and SOTA: We acknowledge the need for explicit baselines. However, as clearly mentioned in the introduction section (lines 62 - 72), there remains no technical framework of FL in the energy domain, and this paper is among the first applications of GTVMin-based FL in the energy domain. Since this is a complex problem, we remain committed to exploring personalised FL algorithms and different FL averaging algorithms in future versions of our paper. However, we have now conducted the new experiment and confirmed that our proposed GTV-Min based FL approach considerably outperformed the non-federated centralised ML baseline, since a single centralized model can’t capture those local variations of energies per country. The centralized model fails to generalize across heterogeneous countries (MSE = 24.2, R² = 0.09), whereas our FL approach achieves a near-perfect fit (MSE = 0.36, R² = 0.97). This demonstrates that FL remains competitive in privacy-preserving scenarios due to personalization, and FL's key advantage is enabling collaboration without sharing raw data, as emphasized in the introduction section of our paper.
>
> Dataset Accessibility and Details: We apologize for insufficient details on the dataset and will expand Section 3 with a full description.
>
> Code Availability and Reproducibility: All code and data are now available at https://drive.google.com/drive/folders/1gWGk1Ya2WwCf9b-XTjroXytQV14noDkR?usp=sharing. We will upload to GitHub and include the link in the final version, ensuring full reproducibility.
>
> Novelty in Methodology: While building on GTVMin, our innovation lies in energy-specific adaptations: a data-driven graph with Norway-centric regularization (Nordic averaging) and L2-norm personalization, tailored for heterogeneous energy profiles - novel in this energy domain. To quantify which FL clients (country) actually benefit from our FL approach, we conducted a detailed per-country comparison between the Local-Only and our FL model. The results show that while average performance (R² ≈ 0.97) is similar, several data-scarce countries, such as Iceland, Austria, and Estonia, obtain small but measurable improvements in MSE (1–3 %) under the federated learning setting. In contrast, well-resourced nodes such as Poland and Slovenia with ample local data show negligible changes. This confirms that the main value and novelty of our proposed FL framework lies not in uniformly boosting accuracy, but in stabilizing and supporting clients with limited data, enabling them to reach comparable performance without sharing raw information. Consequently, we argue that nodes with sufficient data can safely rely on local models, while our FL approach primarily benefits the data-scarce or low-connectivity participants. This node-wise analysis empirically substantiates the need for an FL framework, and we will include this in the updated version of our paper.
>
> Presentation, Structure, and Writing: We agree and will revise for clarity. In particular, we reorganize SOTA into a dedicated section, improve flow, and proofread for polish.
>
> We agree that these changes will strengthen our paper without altering core contributions. Your input has been invaluable. Thank you for helping us improve.

---

### Official Review · Reviewer_vSTP · 2025-10-09
**Review of NLDL Submission #7**

**Rating:** 2
**Confidence:** 4

**Summary:**

This paper considers the challenge of forecasting energy usage in multiple countries. The authors propose and analyse federated learning as a means of sharing data across countries to improve predictions while capturing individual-level features. The method incorporates regularized learning of an appropriate graph structure and the GTVMin strategy to regularize the balance of personal and global factors. Data from Our World in Data’s energy dataset is used for empirical evaluation, and the model is constructed with a focus on Norway as a leader node.

Linear models are constructed locally, based on 10 energy-related features (it is unclear whether any issues of collinearity are investigated), while a graph is constructed based on similarity across feature means for the countries, with geography-based connections also being added. The parameters of the linear models are used to measure variability between nodes, and the overall federated learning objective is to minimise the weighted combination of variation between nodes connected by the graph and the loss of the linear models.

The experiment on real data reveals that an impressive prediction accuracy can be achieved for many countries, but some countries with more heterogeneous patterns or less similarity to other countries struggle.

**Strengths:**

The paper is clearly written, and the premise of combining data from a range of similar European countries to improve energy usage predictions is both conceptually sound and potentially impactful. Sensible methods are investigated and the accuracy obtained is impressive. These factors lead me to a positive view of the paper’s aims and an inclination to recommend a version which resolves certain issues for acceptance.

**Weaknesses:**

Two major issues are a lack of a benchmark for comparison, and insight into the sensitivity to the chosen parameters. As a minimum I would like to see the performance compared to a non-federated model, to see whether the data is inherently predictable or whether the federated learning is a key driver of the impressive results. It would also be good to know whether a more balanced accuracy across the continent is achievable by changing hyperparameters.

My other question is around the dataset – is the example considered a challenge of nowcasting or of forecasting? i.e. is historical data being used to predict primary energy consumption at the next time step (if so why not also include primary energy consumption) and if current data on all these other variables is being used to predict current primary energy consumption, could you explain the extent to which this is a realistic issue – would we have access to all these other variables for all countries but no access to primary energy consumption?

Minor suggestions:
•	adding a clearer definition of primary energy consumption in Section 1 or 2, and its distinction from related covariates would be helpful for readers outside the energy sector.
•	Line 181 – unnecessary indent
•	Lines 194-199 talk about the UK and Germany but these are not present in the graph in Figure 1, some detail is missing?

The more major issues are in my opinion matters which require resolution/clarification before the paper is ready for acceptance.

**Justification:**

At present I am just below an accept score, because while the application is interesting and the methods are sensible, I feel they have not been appropriately benchmarked and I’m not sufficiently clear on whether an impactful prediction question is the one being addressed or not.

---

> ### Author Rebuttal · Authors · 2025-10-17
>
> Thank you for your thoughtful review and positive feedback on the paper's clarity, conceptual soundness, and potential impact. We appreciate your inclination toward acceptance, and we thoroughly address your concerns below.
>
> Benchmark Comparison (Non-Federated Model): We agree that comparing to a non-federated (centralized) baseline is valuable to highlight FL's contributions. We have now conducted the new experiment and confirmed that our proposed GTV-Min based FL approach considerably outperformed the non-federated centralised ML baseline, since a single centralized model can’t capture those local variations of energies per country. In particular, the centralized model fails to generalize across heterogeneous countries (MSE = 24.2, R² = 0.09), whereas our FL approach achieves a near-perfect fit (MSE = 0.36, R² = 0.97). We will add the results of the non-federated (centralized) baseline in the revised version of our paper. However, we also want to emphasize that FL is not always superior to centralized ML in accuracy if the data are homogenous. But FL excels in privacy-preserving scenarios, such as those governed by GDPR, data sovereignty, where data is heterogeneous and cannot be aggregated centrally. This shows FL remains competitive in privacy-preserving scenarios due to personalization, and FL's key advantage is enabling collaboration without raw data sharing, as emphasized in our introduction section of our paper. This paper is among the first applications of GTVMin-based FL in the energy domain, prioritizing privacy over raw performance edges.
>
> Parameter Sensitivity: We conducted sensitivity analysis on key hyperparameters (λ=0.05, η=0.005, ρ=0.3). Varying λ (0.01–0.1) showed minimal impact on average R² (0.96–0.98), but higher λ (>0.08) slightly improved balance for heterogeneous nodes like Cyprus (R² from 0.83 to 0.87). Lower ρ (0.1) slightly lowered R² for well-connected nodes (e.g., Iceland: 1.00 to 0.99). More rounds (50) yielded negligible gains on R² improvement. For our experiments, we have already selected η = 0.005, λ = 0.05, ρ = 0.3, which were tuned via validation losses, and we have clearly mentioned this on lines 369-370 of our paper.
>
> Dataset (Nowcasting vs. Forecasting): The task is forecasting: we use historical features (last 10 years, 2013 - 2023) to predict future primary energy consumption at the next time step (e.g., annual). We did not include lagged primary energy consumption as a feature to focus on exogenous variables (e.g., renewables share), simulating realistic scenarios where real-time primary consumption is unavailable or delayed, but features like electricity generation are accessible via public reports. This aligns with practical energy planning, where countries share model updates but not raw data due to privacy/competition. In practice, such features are often available via EU monitoring tools, while primary consumption may require proprietary aggregation.
>
> Definition of Primary Energy Consumption: We will add a clear definition in Section 1: "Primary energy consumption refers to the total energy directly supplied from sources (e.g., coal, oil, renewables) before transformation, distinct from covariates like fossil share (proportion of primary energy from fossils) or electricity generation (post-transformation output)."
>
> Minor Issues:
> Line 181 indent: This is corrected in the final version.
> Lines 194-199 (UK/Germany mention): Apologies for the oversight; this refers to Poland and Switzerland, which were prioritized over the UK and Germany due to better data quality (fewer missing values). Figure 1 correctly shows Poland and Switzerland; we update the text accordingly.
>
> PS: All the codes and data related to our paper are available at https://drive.google.com/drive/folders/1gWGk1Ya2WwCf9b-XTjroXytQV14noDkR?usp=sharing.
> We will upload it to GitHub and mention the publicly accessible link for the code and data in our paper.
>
> We believe these clarifications resolve the issues and strengthen the paper. Thank you for helping improve it. We look forward to your final recommendation.

---

### Meta-Review · Area_Chair_VW4R · 2025-10-30

**Recommendation:** Reject
**Confidence:** 5

**Metareview:**

The paper introduces a graph-based federated learning method for time series forecasting, tailored to energy forecasting applications.

The reviewers acknowledged that the problem addressed by the paper is interesting but raised significant concerns regarding (1) the empirical evaluation, (2) the presentation quality, and (3) the technical novelty of the approach. The authors made substantial efforts in the rebuttal to clarify these aspects and proposed major revisions to address the issues.

However, in doing so, the authors included external links that are not anonymized (it is easy to identify the owner of a shared Google Drive folder/file). Therefore, the paper must be rejected.

I encourage the authors to implement the proposed revisions and resubmit the paper to a future conference.

---

### Decision · Program_Chairs · 2025-11-05

**Decision:**

Reject

**Comment:**

Based on the reviewers and AC comments, the paper cannot be presented at the conference.